# A Hypothesis: Life Initiated from Two Genes, as Deduced from the RNA World Hypothesis and the Characteristics of Life-Like Systems

**DOI:** 10.3390/life6030029

**Published:** 2016-08-02

**Authors:** Kunio Kawamura

**Affiliations:** Department of Human Environmental Studies, Hiroshima Shudo University, 1-1-1, Ozuka-higashi, Asaminami-ku, Hiroshima 731-3195, Japan; kawamura@shudo-u.ac.jp; Tel.: +81-82-830-1946

**Keywords:** first genes, metabolism, information flow, RNA world, hydrothermal environments, kinetic model, definition of life, Hadean Earth environments

## Abstract

RNA played a central role in the emergence of the first life-like system on primitive Earth since RNA molecules contain both genetic information and catalytic activity. However, there are several drawbacks regarding the RNA world hypothesis. Here, I briefly discuss the feasibility of the RNA world hypothesis to deduce the RNA functions that are essential for forming a life-like system. At the same time, I have conducted a conceptual analysis of the characteristics of biosystems as a useful approach to deduce a realistic life-like system in relation to the definition of life. For instance, an RNA-based life-like system should possess enough stability to resist environmental perturbations, by developing a cell-like compartment, for instance. Here, a conceptual viewpoint is summarized to provide a realistic life-like system that is compatible with the primitive Earth environment and the capabilities of RNA molecules. According to the empirical and conceptual analysis, I propose the hypothesis that the first life-like system could have initiated from only two genes.

## 1. Chemical Evolution and Environments

### 1.1. Introduction: Chemical Evolution

Identifying the molecular structures and functions in organisms, especially in relation to biological information, is a major goal of molecular biology. Biological information is preserved by nucleic acids to direct the biological functions of RNA and proteins through their nucleotide and amino acid sequences, respectively. It is said that the information flow is digitized by associations between the genotype and the phenotype molecules [1,2,3,4]. Furthermore, functional molecules, such as ribozymes and enzymes, induce the formation and breakdown of other biological molecules, and a single enzyme normally possesses a single function for controlling a single biochemical reaction. This idea is strengthened by the fact that information stored by a DNA sequence (genotype) is indirectly assigned a one-to-one correspondence to other molecules (phenotype) downstream (Figure 1) at the molecular level, while there are many quantitative traits affected by many genes. Although the information flow from DNA to peptide sequences of proteins is normally focused on, it should be noted that the formation of other molecules downstream of proteins is also indirectly assigned by DNA sequences. Biological information is transformed from DNA sequences to RNA sequences, peptide sequences and other biomolecules. These functional molecules perform biological reactions, including DNA replication and repair. The cross-links between the information stored in the DNA sequence and the biological functions performed by other biomolecules is a significant characteristic of life-like systems on Earth. The universal machinery for maintaining this sophisticated information flow likely evolved from a common ancestor of our life-like systems, although this is extremely complicated even in the simplest organisms. Most scientists studying the origins of life consider this a key factor in the origin-of-life problem. The fact that this information flow lies in the cross-linked relationship between information and function is one of the most important characteristics of life-like systems on Earth; here, the assignment between genotype and phenotype is extended to that between information and function [4,5]. Thus, it is assumed that the moment when the cross-linked assignment between information and function was established is considered as a great event for the emergence of life-life systems on Earth.

This information flow likely formed slowly, step-by-step, with a number of important chemical evolutionary events, rather than the necessary events happening all at once. Furthermore, it is well established that the relationship between information and function is regarded as the chicken and egg paradox. The RNA world hypothesis is proposed as a hypothesis that would solve this paradox [6], while the idea that RNA would have played the roles of both DNA and proteins was anticipated prior to the appearance of the term “the RNA world hypothesis” [7,8]. The RNA world hypothesis seems to be the most attractive hypothesis for the origin of life; therefore, the validity of the RNA world hypothesis has been extensively evaluated by experimentally simulating the chemical evolution of RNA and in vitro selection techniques of functional RNA molecules. Nevertheless, the RNA world hypothesis still possesses discrepancies from the following viewpoints:
An insufficient number of simulation experiments have been conducted to render the RNA world hypothesis compatible with extreme primitive Earth environments (Figure 2). Especially, the RNA world hypothesis has not been sufficiently evaluated from the viewpoint of the hydrothermal origin of life. This was extensively investigated to verify conditions likely for chemical evolution of RNA and related molecules by using hydrothermal flow reactors by researchers, including our group.There are still unknown pathways for the formation of functional RNA molecules, such as the formation of ribose and the prebiotic replication process.A realistic RNA-based life-like system has not been well described. It is unknown whether the first RNA-based life-like systems may include a quasi-species-type RNA system or a capsulated-type system consisting of several functional RNA molecules.It has not been determined how a simple population of functional RNA molecules became a life-like system that can be regarded as alive. That is to say, the elemental requisites (or characteristics) for defining a life-like system as alive are not completely clear.

Although some of these drawbacks have been evaluated with experimental and theoretical studies, the answers are still unclear. Here, I attempt to draw a picture of the first life-like system based on RNA molecules, considering both the primitive Earth environment and the formulation (and definition) of life.

### 1.2. Answering the Question: Which Was First, Protein or Nucleic Acid?

It is well established that both RNA, protein and their analogs can form under simulated primitive Earth environments. The question of whether nucleic acids or proteins originated first [9] is sometimes discussed. This question is related to the definition of a protein. The term ‘proteins’ involves the molecules formed by organisms. Therefore, this yields a definition of protein that proteins are biomolecules, of which the amino acid sequences are dictated by the corresponding DNA sequences. The fact that replication of peptides is possible under very sophisticated artificial conditions [10] suggests the reproduction of peptides is unlikely without translating from genotype molecules. In other words, the long peptides formed in the primitive Earth environment are not regarded as proteins unless the amino acid sequences of these molecules were assigned by informational molecules. One can assume that the networks consisting of chemical reactions catalyzed by protein-like molecules could form ambiguous information by pseudo-replication [11]. However, experimental and empirical evidence has not validated this. On the contrary, RNA molecules possess both informational and biological functions in vivo and presumably also in simulated primitive Earth environments, although prebiotic replication has not yet been elucidated in entirely aqueous solutions [12], and the RNA world hypothesis is extensively evaluated [13,14,15,16]. In other words, although DNA and RNA can preserve genotypic and phenotypic functions that are coded with sequences of nucleotide residues, protein-like molecules do not have the capability for storing information in this manner [12]. Naturally, protein-like molecules could have been useful to the establishment of an RNA-based life-like system.

### 1.3. Behaviors of Nucleotides and Protein-Like Molecules in Primitive Extreme Earth Environments

It is important to consider the evaluation of primitive Earth environments for the emergence of life-like systems in parallel with chemical evolution experiments, since primitive Earth environments involve a variety of conditions regarding temperature, pressure, pH and atmosphere within the long Hadean history. Presumably, chemical evolution should have occurred between the first evidence of liquid water to 3.8 Ga at most, since the oldest evidence of life is realized at 3.8 Ga [17,18,19], where a suitable environment should have appeared [20]. It should be noted that the present organisms on Earth consist of carbon-based molecules under the mild current conditions where various types of chemical reactions occur. The temperature of the present Earth is regarded to be suitable for chemical reactions for such organic molecules. Although this does not deny that more concrete molecules and materials, such as rocks, or gas phase reactions could form a life-like system, organic molecules in liquid aqueous solutions are normally focused on. Organic reactions in aqueous phases have several advantages, but the formation of life-like systems would be difficult in gas and solid phases. For instance, weak interactions hardly play a role in gas phases, although the possibility of a gaseous form of life cannot be completely ruled out. It would be very difficult to form large complicated molecules in a gaseous environment, especially molecules involving weak interactions, such as hydrophobic interactions and hydrogen bonding. Additionally, ions exist in gas phases only at extremely high temperatures, although ions readily exist in aqueous phases at low temperatures due to ion hydration. Besides, mass transfer chemical reactions in solid phases occur very slowly, with the exception of electron transfer. Thus, organic molecules neither react with other molecules nor form complex associations in the solid phase. The fact that liquid phase environments, and especially aqueous environments, since water is present in the Solar system [21,22], were likely the most suitable phase for life-like systems limits the range of temperatures and pressures for the emergence of life-like systems.

The time range on ancient Earth for the formation of life-like systems is approximately 800 million years, representing a great range of environmental diversity. In addition, geographical variations in environments also exist. For instance, a variety of temperatures would be considered as possible primitive Earth environments. Under the high concentration of carbon dioxide on primitive Earth [23,24], irradiation from the Sun would hardly reach the surface and ocean of primitive Earth. Frequent bombardment by meteorites rose the temperature of the Earth. Besides, luminescence from the Sun was less than the present luminescence [24,25]. Energy sources from the interior of the Earth and the bombardment by meteorites, which control the environment of the Earth’s surface, must be taken into account [24]. Hydrothermal deep-ocean vent systems also played important roles by providing a heat source for biomolecule formation. In these vent systems, high temperature conditions are automatically considered. Presumably, the temperatures of the primitive Earth’s surface may have been fairly high since the Earth started at a high temperature [23]. Thus, a wide range of temperatures, pressures and pH values should be taken into account when studying the chemical evolution of biomolecules.

Second, pH prediction is difficult due to the variety of primitive Earth conditions. The ocean and local hydrothermal vent systems would have exhibited a variety of pH ranges [26,27,28,29]. In addition, the surface of the primitive Earth involved a thick carbon dioxide atmosphere, which would affect the pH of the ocean [26]. Alkaline pH ranges have been evaluated as possible environments for chemical evolution [28]. It is necessary to consider possible primitive Earth environments for chemical evolution, although we do not know the mechanisms by which chemical evolution would proceed under such extreme conditions at high temperatures and pressures with acidic or alkaline medium. A variety of possible environmental factors may have been relevant on primitive Earth other than temperature and pH. Thus, it is important to evaluate chemical evolution under a wide range of simulated experimental conditions. This helps to avoid biased simulation experiments. Thus, establishing an experimental setup that enables investigations under extreme conditions is essential.

Although simulation experiments suggest that both RNA and peptides could have been formed by chemical evolution processes, the efficiencies are normally not very high. In addition, these simulation reactions have been carried out mostly under mild conditions. Phylogenetic trees suggest that the last common ancestor of the present organisms (LUCA) could have been hyperthermophilic [29,30]. Since the LUCA is not the origin of all life, the origin of life at low temperatures cannot be ruled out [14,15,31,32,33,34]. Organic molecules can maintain biological functions at temperatures over 100 °C, since RNA molecules play biological functions at temperatures over 100 °C in modern hyperthermophiles. However, it is important to consider that the hydrothermal origin-of-life hypothesis [32,33,34] would be contradictory to the RNA world hypothesis, when considering whether biomolecules could have accumulated and biomolecules could have expressed biological functions under hydrothermal conditions [14,15,34,35].

Some groups, including ours, have examined the behavior of RNA, peptides and their moieties under extremely high temperatures in aqueous phases by using hydrothermal flow reactors and other unique experimental apparatuses [36,37]. We have developed several types of flow reactor systems, which are useful tools to research chemical evolution under hydrothermal conditions [38,39,40,41,42]. Temperature is a crucial factor, which likely varied on the surface of the primitive Earth. For instance, the reaction rates of RNA-mediated processes are enhanced 2–3-times with every 10 °C increase in temperature [39,43,44]. The range of temperatures where organisms could survive is indeed wide. Pressure is also important for reaction behavior, since deep ocean and crust environments are candidate locations for the chemical evolution of biomolecules [45,46]. According to the results of quantitative experiments in our hydrothermal flow reactor systems, the stability of these molecules decreases with increasing temperatures. The stability of RNA is lower than that of peptides [39,43,44,47,48]. However, this is a relative evaluation, and the time scale of degradation is frequently evaluated from the human perspective. The shortest half-life of these molecules is on the millisecond scale, even at extreme temperatures around 300 °C. This rate corresponds to the reaction rate of modern enzymes [15,34,49]. Our results demonstrated that RNA molecules could have accumulated even under hydrothermal conditions since their accumulation is determined by both the formation and degradation of RNA. When comparing the formation and degradation of oligonucleotides, results suggested that the accumulation of RNA molecules would indeed be possible at temperatures over 300 °C. This suggests that the accumulation of biomolecules in a life-like system is kinetically possible under the conditions where energy and materials inflow to and outflow from the system. Thus, the stability of biomolecules should be evaluated from this kinetic viewpoint. The concentrations of biomolecules are indeed regulated by both the formation and degradation rates.

In addition, three-dimensional structures of biomolecules have been evaluated at high temperatures beyond 100 °C. However, this does not necessarily suggest that the RNA world hypothesis is incompatible with possible extreme Earth environments. This was also observed using our hydrothermal flow reactor with in situ observation of UV-visible absorption spectra. Although biologically-important weak interactions, such as hydrophobic interactions and hydrogen bonding, become weaker with increasing temperatures, such interactions can still occur even at high temperatures [41]. On the other hand, our studies showed that the solubility of these molecules at high temperatures is another important factor for evaluating the chemical evolution of biomolecules [50,51]. According to these data, the RNA world and hydrothermal origin-of-life hypotheses are possibly compatible, where a life-like system consisting of organic molecules would be possible even at very high temperatures. The following three important factors are summarized for the expression of biological functions by biomolecules at high temperatures:
The relative rates of the formation and degradation of biomolecules, such as phosphodiester bonds of RNA and amide bonds of peptides.The biologically-important weak interactions by hydrogen bonding and hydrophobic interaction.The solubility of these molecules.

## 2. Life-Like Systems Deduced from the Definition of Life

### 2.1. Problems Regarding the Definition of Life

Studies including evolutionary engineering of functional RNA molecules under simulated Earth conditions to construct life-like systems are a major approach for investigating the origin of life, which is supported by empirical evidence on primitive Earth environments. Alternatively, tracing back from modern organisms to ancient life-like systems is another approach [29,30,31,32,33,34,35]. Although these two directions of research, from chemistry to biology and biology to chemistry, may merge to find the origin of life, there is still a great gap between the two approaches.

Analyzing the definition and characteristics of life would be a third approach for solving the origin-of-life problem. In other words, identifying the origin of life is almost equated to clarifying the definition of life. However, it is very difficult to create a definition that most scientists agree upon [52,53,54]. In the present study, I analyze the definitions of life to deduce a realistic picture of a life-like system consisting of RNA molecules. The definitions of life have been summarized in the following viewpoints:
The first step is to focus the characteristics of life-like systems into life on Earth.The characteristics that life consists of cell(s), metabolism, replication and evolution are well accepted.To maintain these characteristics, the fact that life possesses machinery for the replication of information and the transformation of that information to accomplish biological functions is also accepted.

### 2.2. Approaches for Defining Life

There are two main approaches to create a definition of life (Figure 3). One is to dissect characteristics of life-like systems into undividable elemental characteristics. This would deduce a picture of life that possesses these elemental characteristics. This may be regarded as reductionism. The characteristics of metabolism, amplification and evolution can be regarded from this direction of definition.

On the other hand, a definition of life can be achieved on the basis of holism, although it seems that evaluating the definition of life is not usually discussed from this viewpoint. I propose that the role of a life-like system, such as a cell-based organism, in relation to the environment can be one approach to create a definition of life (Figure 4), which was demonstrated in my previous work [2,3,4]. In other words, life-like systems show the trend that building blocks do not normally interact directly with the environment when a biosystem also interacts with the environment. According to this principle, I proposed a hypothesis that the machinery, which supports evolution, is also regarded as a strategy formed in life-like systems to adapt to environmental changes [2,3,4]. Although one may regard this nature as autonomous or self-contained, the term autonomy does not describe the relationship between life and its environment. Thus, if a life-like system can be regarded as alive, an autonomic and self-contained nature towards the environment becomes clear. Naturally, we need to adopt this nature for RNA-based life-like systems, if RNA-based life-like systems were indeed the first organisms.

Here, a prejudice in relation to Darwinian evolution should be noted. The original Darwinian evolutionary theory consists of mutations and natural selection. However, it would be a somewhat narrow view if mutations are viewed as the only mechanism of introducing new functions into organisms. For instance, sexual reproduction involving meiosis is an inherent method of shuffling genes in eukaryotic species. If this type of mechanism were regarded as a product of natural selection, it would be true that the interaction of life with its environment is considered a requisite of organisms. Recently, it was discovered that some epigenetic influences beyond simple genetic mutations appear to be heritable [55]. This idea agrees with an autonomic or a self-contained nature of life rather than a special vital nature in organisms.

### 2.3. Comparative Analysis Using Analogies among Biosystems as a New Approach for Characterizing Life

I proposed a new approach to compare biosystems at different hierarchical levels to identify the characteristics of life-like systems [4]. There are several types of life-like systems at different hierarchical levels in biota, including eukaryotes, prokaryotes, social insects, ecosystems, species, civilizations, viruses and viroids. Although single-celled (prokaryotes and eukaryotes) and multicellular organisms (eukaryotes) are readily accepted as cell-based organisms, some life-like systems, such as species, societies of social insects, ecosystems and civilizations, possess somewhat different characteristics from the cell-based life-like systems in relation to the role of the system and its building blocks for the environment (Table 1). The characteristics of life-like systems, including cell-based organisms, were analyzed in my previous publication [4]. As a result, these biosystems can be classified into a broad spectrum of life-like systems, depending on which informational controlling machinery is involved at the hierarchical level of the system, beyond the hierarchical level of its elements. Depending on whether the biosystem possesses inherent machinery to replicate (or amplify), modify, shuffle or incorporate new information into the system, these biosystems can be classified into some categories. Analogies among biosystems at different hierarchical levels deduced the importance of replicating information, the assignment between information and function and the incorporation of new functions into the system. In general, this is named as the central controlling system for information (CCSI) (Figure 5). The nature of CCSI was briefly described in my previous publication [56,57,58,59].

At the same time, metabolism, especially controlling the supply of energy and materials into the system, is to be dictated by the CCSI; this is named as the central controlling machinery for the inflow/outflow and formation/degradation of energy, materials and information from environments (CMIO) (Figure 5). CMIO is essential to support CCSI, and CCSI directs the CMIO machinery. In other words, CMIO may be regarded as an extended term for “metabolism”. The CMIO of modern organisms is involved in several metabolic pathways. This relationship is similar to the assignment between information and function, between genotype and phenotype and between DNA and RNA or protein (Figure 6). Conclusively, establishing CCSI is not sufficient to make a system ‘alive’. This viewpoint is consistent with the fact that organisms can be regarded as dissipative structures under non-equilibrium thermodynamics, and metabolism is regarded as an essential requisite for a system to be considered ‘alive’.

Here, it is pointed out that organisms possess a universal information flow at the basic level, but differences and similarities among organisms at different hierarchical levels should be focused on in detail. First, the differences between prokaryotes and eukaryotes are clarified on the basis of the viewpoint that the replication and transformation of information of DNA to functional molecules seems to be common. Genetic information stored in DNA is transformed into functional molecules, and the functional molecules support maintaining CCSI itself. These biosystems involve common characteristics of the codon table, similar structures of tRNA and rRNA, as well as other similarities (Table 2). Although most materials involve the same type of molecules for controlling replication, information flow, etc., in eukaryotes and prokaryotes, inherent prokaryotic machinery, which is not present in eukaryotes, can be identified at different hierarchical levels of organisms. The machinery for replication and transformation in eukaryotes is indeed separately carried out inside and outside the nucleus, while the corresponding prokaryotic machinery is floating in the cell. Second, the CMIO machinery is also much different between prokaryotes and eukaryotes. Eukaryotes, especially multicellular organisms, normally possess inherent structures for obtaining materials and energy from their environment. The formation of eukaryotes is largely supported by the emergence of mitochondria, which are not present in prokaryotes. These cellular organelles provide almost the entire energy supply for eukaryotes. Additionally, eukaryotes possess inherent machinery for reading DNA sequences beyond the boundary of the cell nucleus, which is not present in prokaryotes. Thus, prokaryotes and eukaryotes are considered to be quite different biosystems from both CCSI and CMIO viewpoints. As another example, multicellular organisms possess different types of machinery to mix (shuffle) genes at their inherent hierarchical level as a system. Nervous systems and hormones are also inherent regulatory mechanisms that are not present in single-celled organisms. The third example shows that the metabolism corresponding to CMIO is maintained by organs formed by cell differentiation; therefore, the formation of multicellular organisms is supported by cellular differentiation. This established the machinery of energy and material supply for multicellular organisms as biosystems.

Similarly, for the system of social insects, the society is regarded as a population of individual insects, but is also linked with its specific machinery to produce descendants. This machinery can be regarded as an inherent CCSI mechanism for maintaining the social insect biosystem, beyond the hierarchical level of the individual workers as building blocks. The formation of a system of social insects is supported by the differentiation of individuals who support the energy and material supply for the system in different ways, as members of the insect society. Thus, it is regarded that the energy supply is maintained by the inherent machinery of CMIO at the societal level.

On the contrary, ecosystems do not possess inherent mechanisms corresponding to CCSI and CMIO. However, civilizations, which are the products of human societies in history, possess their own machinery for replication (amplification), transformation of information to functions and assignment between information and function [2,3,4]. I previously emphasized that civilizations are a unique system involving CCSI and CMIO by inherent mechanisms, while social groups of organisms, including even non-civilized human societies, do not necessarily possess their own inherent CCSI and CMIO at the social level. The energy and material supply of early and modern civilizations were supported by agriculture and fossil fuels, respectively. This principle was successfully applied to describe human history [57,60].

Conclusively, the biosystems of cellular organisms, social insects and civilizations possess inherent machinery for both CCSI and CMIO, although it is sometimes difficult to determine that social insects and civilizations are considered as alive at a higher hierarchical level. The present paper demonstrated a hypothesis that CCSI and CMIO are essential functions for a system to be regarded as alive, for the first time, on the basis of biosystems analogies at different hierarchical levels. Thus, the biosystems can be classified into roughly two categories. One is biosystems that have inherent CCSI and CMIO, and the other one is biosystems that do not have clearly inherent CCSI and CMIO. According to this discussion, both CCSI and CMIO are essential for the emergence of a biosystem beyond the level of its building blocks, as shown in Table 3. CCSI consists of four main functions. CMIO, which is directed by CCSI, provides energy, material and information from the environment to support CCSI. Characteristics 1–4 concerning CCSI and Characteristic 5 concerning CMIO were considered the essential functions for drawing the elemental requisites, which construct biosystems, for life. Acquisition of the CMIO function by a biosystem for energy and material supply is an event that turns a simple population of building blocks into a biosystem at a higher hierarchical level.

An important point regarding CCSI and CMIO should be noted, that major transitions of biosystems have happened only a few times during the long history of life-like systems on Earth; that is, the emergence of the first cell-type organisms from a simple population of biomolecules, the evolution of eukaryotes from prokaryotes, the evolution of multicellular organisms from single-celled organisms, the evolution of social systems of individuals, including social insects, and the evolution of civilization from human society. Comparing these biosystems in detail would be an effective approach for determining the emergence of the simplest biosystems, that is the first cell-type organisms. That is to say, comparing several types of biosystems with inherent CCSI and CMIO could deduce a general rule for the emergence of a life-like system or the transformation of a life-like system to a higher hierarchical level possessing both CCSI and CMIO beyond its individual elements. It is assumed that this transition occurs only in the case that the cross-linked relationship is established between CCSI and CMIO.

## 3. Estimating the Most Primitive Life-Like System

### 3.1. Importance of Energy and Material Supply by CMIO

The relationship between CCSI and CMIO is similar to the assignment between information (genotype or DNA) and function (phenotype or RNA and protein). This indicates that there is an alternative “chicken or egg problem” regarding the origin of life other than the “chicken or egg problem” regarding DNA and protein (Figure 6). Here, I attempt to clarify the importance of CCSI and CMIO as general characteristics of life-like systems. Based on the viewpoint described in the present paper, if RNA-based life-like systems appeared as the most primitive organisms, the machinery for CCSI and CMIO should have been present in this system, where RNA molecules should have played central roles for the emergence of both CCSI and CMIO. This would be much simpler than the system for present organisms, since the functions of both DNA and protein were maintained by RNA molecules. The emergence of CCSI would be possible if we assume that the replication of information was maintained by RNA replication and the incorporation of new functions into the organisms was performed through errors during RNA replication. On the other hand, the emergence of CMIO, which is considered as the origin of metabolism, should have been also maintained primarily by RNA molecules.

### 3.2. A Picture of the Most Primitive Life-Like System Deduced from the Biosystems Analogy

The formation of CCSI and CMIO is essential for the emergence of biosystems beyond the hierarchical level of their elements, where the cross-linkage between CCSI and CMIO would be essential for the emergence of life-like systems (Figure 5). Here, it should be noted that the elements of eukaryotes and civilizations are organisms and the elements of prokaryotes are molecules. Thus, one may consider that the behaviors of eukaryotes and civilizations might be attributed to the fact that their building blocks are organisms. On the other hand, there is a trend that a system’s building blocks lose their life-like characteristics and behave as just building blocks of the system for the higher hierarchical level when the system is behaving as a life-like system [4]. For instance, individual cells of a multicellular organism do not interact with the environment directly, but behave as parts of the biosystem at the upper hierarchical level. Thus, the emergence of CCSI and CMIO from non-living building blocks is not in conflict with this principle.

A working hypothesis at the next stage is to estimate how such a system consisting of CCSI and CMIO could be formed from the simple gathering of prebiotic molecules. Compartments (cells) would be a strong strategy, making CCSI and CMIO a spatial unit of the biosystem. RNA molecules without capsulation would have hardly encountered other molecules, since the transportation of such molecules under primitive Earth environments was dependent on diffusion. Otherwise, chemical evolution in an isolated small pond containing RNA molecules would be suitable to avoid extreme dilution.

The size of cells in modern organisms is indeed suitable to control reactions only by diffusion. In other words, the reaction rates are not readily affected by other physical conditions, such as mixing and flow. It is readily assumed that cell-type systems are suitable for avoiding such difficulty. However, we need to search other possible methods where molecules meet other molecules within a sufficiently short time. If the size of the cell is assumed hypothetically to decrease down to a minimal size, it finally reaches the style of viruses, where the inner aqueous medium of the cell is miniscule to nonexistent. The average size of a prokaryote is 1–10 μm in diameter, with the minimum size of a cell being 0.5 μm in diameter, which is roughly 1000-fold greater than the diameter of a water molecule and 10–100-fold greater than the size of typical viruses. Modern life-like biosystems, excluding cell-type systems, such as viruses and viroids, possess only incomplete CCSI and CMIO. It is generally said that modern viruses and viroids do not possess autonomous CCSI and CMIO, although modern huge-scale viruses may have developed functions that are more sophisticated [61]. A reason that viruses could not possess CCSI and CMIO may be because viruses do not possess sufficient inner volume to store water molecules. Thus, the minimum size of the first life-like systems would be determined by the number of molecules needed to maintain CCSI and CMIO.

### 3.3. Two Essential Genes Initiated a Primitive Life-Like System

How many genes or functions were necessary and how were CCSI and CMIO systems constructed for the emergence of life? If we propose a number of the first essential genes and their functions, this hypothesis can be evaluated empirically. Now, the following discussion will deduce that two genes were essential for the construction of both CCSI and CMIO, where a single gene was required for each process. For the formation of CCSI, the establishment of replicase function should be essential on the basis of the RNA world hypothesis, where RNA molecules could have replicated with an RNA polymerase ribozyme consisting of an RNA molecule [62,63]. If the replicase function involves errors, the CCSI involves mutations. This satisfies Functions 1–4 for CCSI.

On the other hand, in modern organisms, the supply and consumption of energy and materials are regarded as a circular network, including the formation (or incorporation) of activated monomers and the degradation (or discharge) of reaction products. The overall network is maintained by a series of enzymatic reactions in organisms. A cell-based organism contributes to a part of the circular network in modern biota. It is indeed difficult to distinguish the border of the circular network since all organisms and all ecosystems link to each other more or less on the Earth. The biological processes involved in the circular network occur at very high reaction rates, and the reaction rates are adjusted to a narrow range of reaction rates by enzymes in modern metabolic processes [15,49,64,65]. If one traces back this trend to the origin of life, the elemental reaction processes had very slow rates without enzymes (or ribozymes) and a large range of reaction rates. To link this non-catalyzed network with CCSI, a single gene for the construction of CMIO was necessary to accelerate the slowest process within the non-catalyzed circular network. This principle is illustrated in Figure 7. This resembles the fact that modern metabolism involves circular networks, such as the TCA cycle, and since life-like systems are only capable of emerging under non-equilibrium environments, the total mass is constant. The contribution of life-like systems should be dependent on with and without a compartment. For the case of a cell-based system with a compartment, each life-like system is capable of contributing to a part of the reactions in the network maintained inside of a cell. For the case of non-cell-based systems, such as a population of RNA molecules in a closed system without a cell-based compartment, where each RNA molecule is regarded as an individual element [66], the whole of the processes in the network may be considered. This model is analogous to an illness where a particular metabolic pathway is disabled by the lack of an enzyme. The circular network can be indeed disconnected by the lack of a single enzyme, which can be caused by a single mutation. In other words, the overall circular network is blocked by a slow process due to the lack of an enzyme.

This principle can be traced back to the initial stages where the evolution of primitive circular networks occurred by mutations in CCSI. In other words, there were a large number of possible processes, which would have been potentially accelerated by mutations in CCSI. Arrows indicate the evolution of the circular network (Figure 7), where the circular network in relation to RNA molecule metabolism proceeds with very slow reaction rates in the absence of ribozymes. Naturally, CMIO could not have evolved unless a cross-linkage between CCSI and CMIO was established. This is also illustrated in Figure 8, where the processes for the formation and degradation of RNA molecules were accelerated by newly-emerged ribozymes assigned by CCSI. The circular network indicates a simplification of a large number of possible degradation pathways going back to the formation of an activated monomer and oligomer, where the first life-like system contributed to accelerate a part of the processes. In other words, it is not necessary that a most primitive life-like system include the whole circular network. This should be related to the style of compartment for these reaction processes. At the same time, reactions without ribozymes possess different reaction rates. This means that the slowest process determines the overall rate of the circular network. This is analogous to the case mentioned above for the illness caused by a single mutation. Thus, if the slowest process were accelerated by a newly-emerged ribozyme from a random mutation within CCSI, the overall rate of the circular network was increased. That is to say, the time when this gene emerged is regarded as the time when the interaction between CCSI and CMIO began. In other words, the construction of CMIO was initiated by CCSI. Conclusively, this evaluation implies that the emergence of a single gene was necessary for the initiation of a life-like system.

Once the slowest process was accelerated and the slowest reaction rate became faster than the second-slowest process, so the second-slowest pathway became the limiting step in the overall circulation process (Figure 7). Thus, the overall rate of the circular network will not be enhanced unless the second-slowest process is catalyzed by the emergence of a second ribozyme, from which a second new gene emerged by a random mutation in CCSI. Repetitive step-by-step accelerations of this circular pathway by gene mutations in CCSI could have improved the whole metabolic circular network; this is illustrated as the narrow arrows, indicating that slow rates are gradually replaced with the bold arrows indicating fast rates. This principle can also explain the reason why modern enzymatic reaction rates are controlled at a narrow range of reaction rates as mentioned above [64]. Conclusively, slow processes can be consecutively replaced by fast processes with ribozymes by consecutive CCSI mutations. This may be analogous to protein metabolism in relation to nutrition, where the total amount of protein synthesis is restricted by the smallest amount of essential amino acids.

The restriction that the reaction rates of all processes were absolutely equal is not essential since reaction rates are altered readily by different concentrations of ribozymes and other reaction conditions. Thus, only a mild restriction that reaction rates were roughly the same is important.

### 3.4. Could RNA-Based Life-Like Systems Form Minimal Life-Like Systems?

Common images of the RNA world can be classified into two categories. One idea proposed by Eigen’s pseudo-species indicates that RNA molecules correspond to individuals, where individual RNA molecules are subject to Darwinian evolution. As I pointed out above, RNA molecules only rarely interacted with other RNA molecules in this type of system, unless the populations of RNA molecules are confined in a small space with natural boundaries, such as in a small pond or inside minerals. Naturally, the second idea is that a cell-like compartment including a number of RNA molecules would be a simple strategy, which seems to readily translate into the emergence of prokaryotes. According to the above analysis, CCSI and CMIO are necessary as minimum functions for life-like systems. Therefore, the question is whether the essential part of CCSI and CMIO could be constructed by mainly RNA molecules. It can be assumed that the primitive replicase ribozyme with mutation errors is capable of satisfying Functions 1–4 for constructing CCSI. It is indeed demonstrated that RNA polymerase ribozymes can be evolved by artificial in vitro selection [62,67,68]. However, as I pointed out in my previous work, the replication conditions by entirely prebiotic materials are not yet established, although most RNA world scientists believe that RNA can replicate under simulated primitive Earth environments in an aqueous medium [14,68].

There is a debate about whether CMIO could have formed by a CCSI consisting of entirely or mainly RNA molecules. This idea has not been extensively investigated [69,70,71]. The most important type of energy supply for RNA-based life-like systems would be activated RNA monomers or an energy supply to make activated RNA monomers, such as condensation agents. Processes that ribozymes use to catalyze main pathways, which link RNA moieties with the formation of activated RNA monomers, would have been an essential step at the final stage of evolution after the initial linkage between CCSI and CMIO (Figure 8). Such processes include the formation of bases, ribose, nucleosides, nucleotides, activated nucleotides and activating agents for these pathways. A similar idea was derived from a different concept, which includes autocatalytic feedback [72,73], which might be related to the present idea. As mentioned in the previous section, acceleration of the slowest process by a ribozyme initiates the following enhancement of the overall rate of the circular network. Naturally, it is readily assumed that a process as a building block of CMIO would be enhanced by a protein-like molecule, of which the formation of the protein-like molecule was enhanced by a ribozyme. This ribozyme can be regarded as a composing element of CMIO. In addition, a cell-type compartment would be strong strategy for stabilizing the life-like system. Thus, a key gene would have initiated the establishment of a compartment to improve a system consisting of CMIO and CCSI on the basis of the same principle for the establishment of CMIO shown in Figure 7.

The first life-like system after the linkage between CCSI and CMIO would have been supported mainly by resources formed from spontaneous chemical evolution. This means that the materials concerning RNA metabolism shown in Figure 8 should have been somewhat accumulated by chemical evolution prior to the linkage between CCSI and CMIO. For completing the circular network controlled entirely by ribozymes, for instance, the accumulation of hydrolyzed RNA products should have been solved by the emergence of particular ribozymes. For instance, 5′-modified nucleotides with triphosphates or other activated groups would be the primitive activated nucleotide monomers, but the hydrolyzed products after oligonucleotide digestion are normally 2′- or 3′-phosphates of nucleosides. Thus, the conversion of 5′-modified nucleotides to 3′-modified nucleotides should consist of many processes, which would have been essential once CMIO was initiated. Thus, it is important to identify which was the slowest process among the circular network regarding the primitive RNA metabolism as a future subject.

This hypothesis also implies the size of a minimal life-like system. If only two genes were essential for the initiation of CCSI and CMIO, the size of the initial genes could be very small. Based on this principle, the minimum size of an RNA-based life-like system can be estimated. If each function corresponds to a size of 100 base pairs for each gene as an example, the total genome size would be 200 base pairs owing to the presence of two genes; a base nonspecific RNA polymerase ribozyme consisting of only 100 nucleotides has not yet been discovered [62,67,74]. The size of an RNA molecule with 200 bases would be approximately 4 nm × 4 nm × 20 nm in size. On the other hand, a top-down approach observing the smallest modern organisms could not readily narrow down the number of functions necessary for a life-like system to lower than a few hundred [75,76]. Naturally, it is hard to imagine that such a large number of functions appeared at one time. On the contrary, the present hypothesis proposes that the emergence of only two genes is capable of initiating step-by-step evolution. 

One should remember that the present hypothesis was deduced from the comparative analysis of analogies among biosystems at different hierarchical levels, where the establishment of CCSI and CMIO results in the formation of a biosystem at a greater hierarchical level beyond the system’s building blocks. For instance, the establishment of CMIO in the first life-like system is analogous to the fact that the establishment of large-scale irrigated agriculture is an essential step for the formation of a civilization, which possesses its inherent CMIO and controls the agricultural system. In this case, human brains were used to maintain the primitive inherent CCSI at the upper hierarchical level. This analogy also reinforces the importance of the interaction between CCSI and CMIO.

Conclusively, the present discussion demonstrated the importance of cross-linked CCSI and CMIO for the emergence of the most primitive life-like system, where a single gene was essential for CCSI formation and another single gene was essential for CMIO formation. In addition, the proposed principle is consistent with the idea that non-life and life are continuous. 

## 4. Conclusions

The present paper demonstrated the importance of a comparative analysis using analogies among biosystems at different hierarchical levels, where the formation of cross-linked CCSI and CMIO is an essential step for the emergence of a biosystem at a higher hierarchical level for chemistry. These analyses enable us to deduce that the emergence of cross-linked CCSI and CMIO was essential for the formation of the first life-like system. In addition, only two genes were essential for the cross-linkage between CCSI and CMIO, and this cross-linkage would have promoted the continuous formation of new genes. In other words, this could have initiated further continuous evolution of the life-like system by mutations in CCSI under Darwinian evolution pressure. This principle fits the definition of life by NASA, that it is a self-sustained chemical system capable of undergoing Darwinian evolution. Thus, this paper discusses the hypothesis that the generation of two genes was the first step for the emergence of life.

## Figures and Tables

**Figure 1 life-06-00029-f001:**
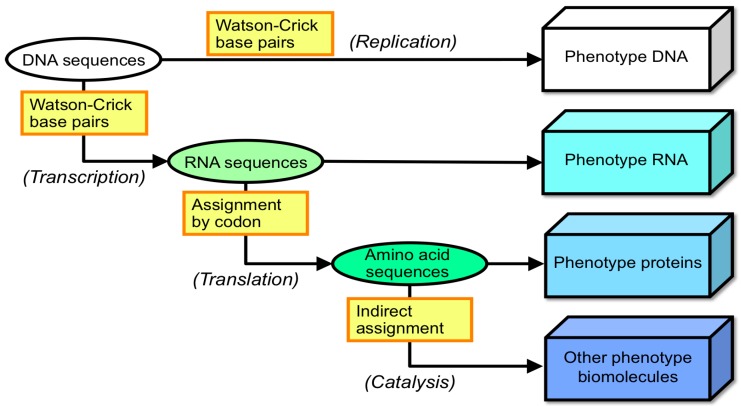
Assignment flow of biological information to functional biomolecules downstream. Extended information flow is illustrated beyond the central dogma. RNA and proteins are phenotype molecules, but also possess roles for assigning other biomolecules downstream. From this viewpoint, DNA molecules are also regarded as phenotypes, which would possess only a function for replication.

**Figure 2 life-06-00029-f002:**
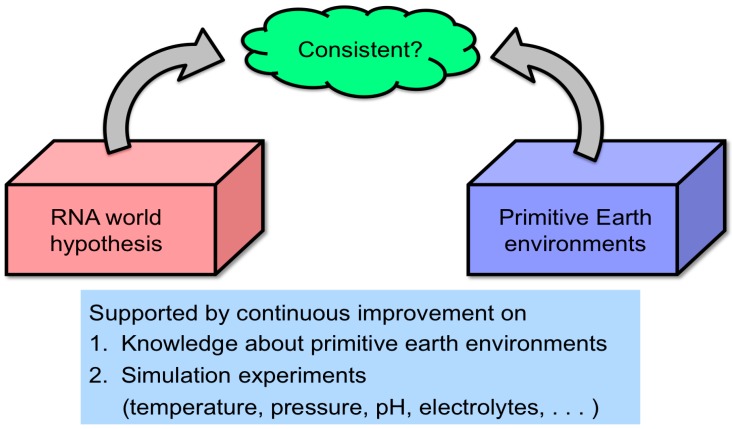
Since the RNA molecules preserve both biological information and functions, the RNA world hypothesis has been extensively investigated. However, continuous studies on geochemistry and hyperthermophiles suggest a diversity of primitive Earth environment, especially the hydrothermal environment. Thus, the RNA world hypothesis should be reconsidered to be compatible with the ancient Earth environments.

**Figure 3 life-06-00029-f003:**
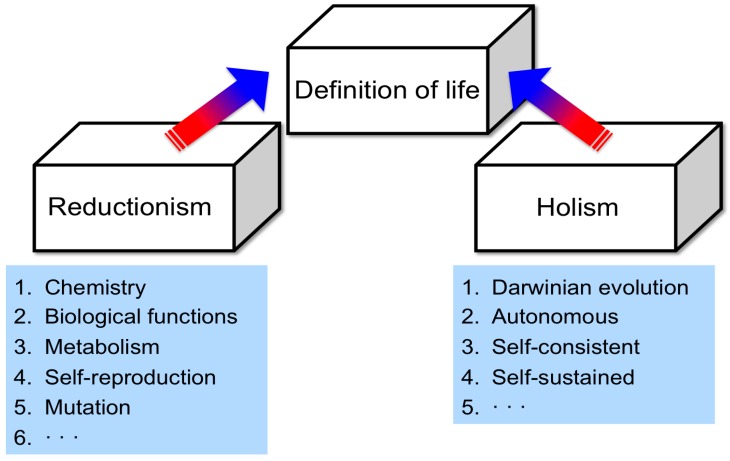
Definition of life as an important approach to deduce the origin of life. The definition may be categorized from reductionism and holism.

**Figure 4 life-06-00029-f004:**
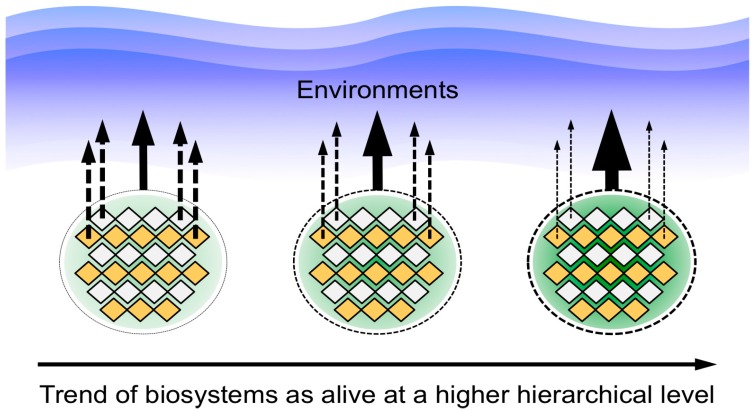
The role of the building blocks and the biosystem in relation to their environment is a unique and useful viewpoint for the definition of life. A general trend that building blocks do not interact directly with the environment when the biosystem at a higher hierarchical level does interact directly with the environment is illustrated with narrow and bold arrows. The illustration was drawn with modification on the basis of the principle shown in publications [2,3,4].

**Figure 5 life-06-00029-f005:**
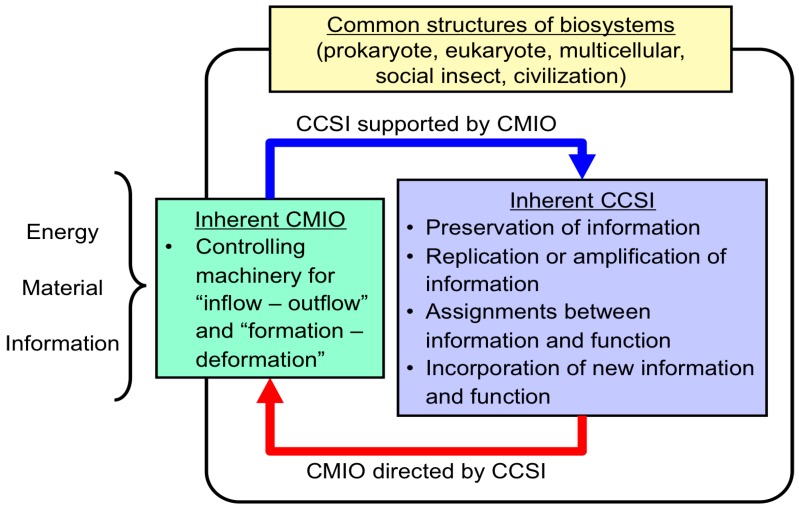
Common structures among biosystems, including prokaryotes, eukaryotes, social insects and civilizations, possess both inherent machinery for the central controlling system for information (CCSI) and the central controlling machinery for inflow/outflow and formation/degradation of energy, material and information from environments (CMIO). CCSI and CMIO construct a cross-linkage. The illustration was drawn with modification on the basis of the principle shown in publications [56,57].

**Figure 6 life-06-00029-f006:**
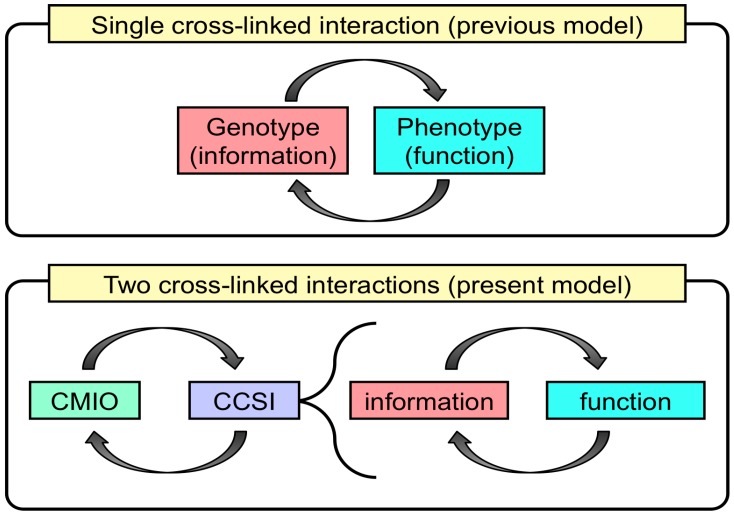
Importance of another “chicken or egg problem” between CCSI and CMIO beyond the relationship between “genotype and phenotype”. CMIO is regarded as an extended term for “metabolism”, which is assigned by CCSI.

**Figure 7 life-06-00029-f007:**
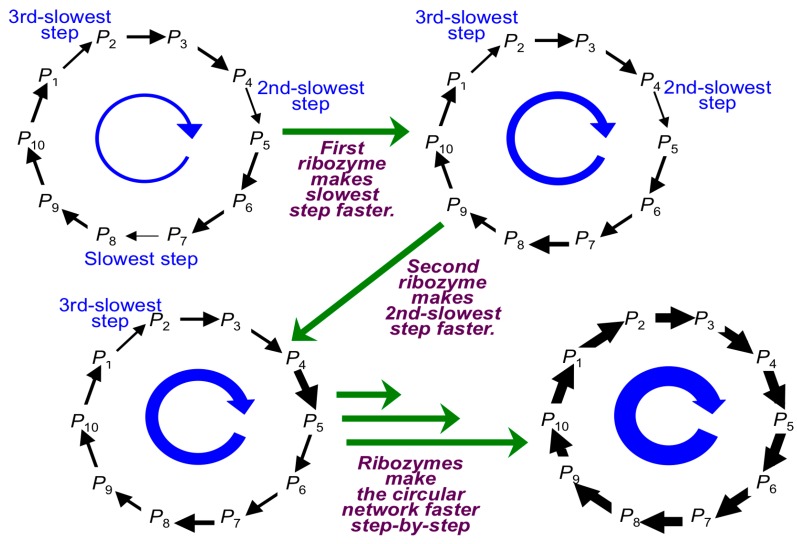
Stepwise improvement of a circular network of CMIO. The overall rate of the circular network was very slow with different reaction rates, where the slowest rate limits the overall rate of the circular network. If the slowest process were accelerated where the slowest reaction became faster than the second-slowest process, the second-slowest pathway became the limit of the overall rate of the network. Thus, the second-slowest process will be the next target, for which a second new gene emerging by a random mutation in CCSI would have accelerated the second-slowest reaction rate faster than other reaction processes. Repetitive step-by-step accelerations in the circular network by gene mutations in CCSI could have improved the whole metabolic circular processes, which gradually replaced the narrow arrows, indicating slow rates with the bold arrows indicating fast rates.

**Figure 8 life-06-00029-f008:**
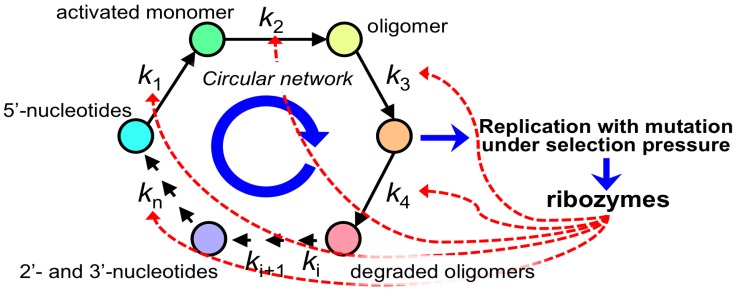
The principle of the step-by-step evolution of CMIO directed by CCSI exemplified for the pathway of the metabolism of RNA molecules. If the slowest process were enhanced by a ribozyme, the circular pathway is accelerated by CCSI. This event supports the CCSI.

**Table 1 life-06-00029-t001:** Biosystems at different hierarchical levels and attributes used for comparative analysis of biosystems.

Biosystems at Different Hierarchical Levels	Attributes
● RNA-based life-like system (RNA world)	● Boundary
● Prokaryote	● Metabolism
● Eukaryote (single celled)	● Information
● Multicellular organism	● Preservation of information
● Social insect	● Self-reproduction
● Society of organism	● Assignment between information and function
● Ecosystem	● Incorporation of new information and function
● Civilization	● Role of biosystem and its building blocks in relation to their environment

**Table 2 life-06-00029-t002:** Inherent machinery of CCSI and CMIO for biosystems at different hierarchical levels.

Biosystems	Inherent CCSI	Inherent CMIO
Prokaryote	The information flow from DNA, RNA, proteins and indirectly to other molecules, by transcription, translation and by enzymatic catalysis	Energy and material flow, including information through membrane proteins
Eukaryote (single celled)	Genetic information is withdrawn through nucleus and used in cytoplasm	Subcellular organelles, such as oral groove, gullet and food vacuole, are used for the incorporation of materials
Multicellular	Mixing of genes by means of sex for reproducing next generations	Organs, such as mouth, root and leaves, are used for gaining material and energy
Social insect	Differentiation of queen, worker, soldier, etc., for reproducing next generations	Workers gather to stock materials for the society of individuals
Civilization	Social systems and instruments for education and research	Social systems and instruments for production and consumption

**Table 3 life-06-00029-t003:** Functions regarding CCSI and CMIO.

CCSI supported by CMIO
1. Preservation of information
2. Replication (or amplification) of information
3. Assignment between information and function
4. Incorporation of new function linked with its information into the biosystem
**CMIO directed by CCSI**
5. Central controlling machinery for inflow and outflow of energy, material and information from environments

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
