# Peer review of "A Hypothesis: Life Initiated from Two Genes, as Deduced from the RNA World Hypothesis and the Characteristics of Life-Like Systems"

_life, 2016, doi:10.3390/life6030029_

Round 1

Reviewer 1 Report

line 38 - The linking of 'other molecules' to 'phenotype' here is a little odd. There may be a link from one gene to one protein, but one protein is not really a phenotype. There are many quantitative traits affected by many genes.

line 73 - the point is made that current RNA World theories do not seem compatible with a hydrothermal origin of life. But it is not clear where this leads us. Maybe life did not begin with RNA. Maybe life did not begin with hydrothermal vents. Maybe there is some slightly different environment that we would still call hydrothermal in which RNA is sufficiently stable. Maybe there is some slightly different nucleic acid-like polymer that is stable in hydrothermal conditions. Too many unknowns here to be a useful point.

lines 93-106 - I mostly agree with this paragraph but it is not said very clearly. Certainly we need to distinguish between long peptides formed spontaneously and those that are translated. But several questions still remain - were there long peptides before long RNA strands? Was there a way of reproducibly making the same amino acid sequence without translating it from RNA? If not, then is there any way that random non-encoded peptides could be useful to RNAs?

The idea in caption to Fig 4 that building blocks do not interact directly with the environment seems interesting and plausible, but not very well defined. Are you saying a cell interacts with the environment but a single gene does not? Or a multicellular organism interacts but one cell does not? The citations 52 and 53 are in fields that are not related to biology and origin of life. So I think this idea needs to be explained and justified in a biological context.

Fig 5 - drawing parallels between systems of different levels of complexity (from prokaryotes up to civilizations) is interesting, but it is a bit distracting at this point. The abstract promised to talk about RNA World and the origin of life. At this point the article seems to have strayed a long way from this intention. From the origin of life viewpoint, even the simplest of these (the prokaryote) is still very complex. The issue for the origin of life is how to get to a prokaryote. I don't really think that looking at social insects and human societies will help much in understanding the origin of life. The ideas of CCSI and CMIO seem interesting, but they are discussed with the high-level examples (pages 10-11), not with molecular and cellular examples. Probably a lot of this could be simplified. It is only when we get to paragraph 3.1 that we get to the point of the article.

Fig 7 - Viewing a metabolism as a cycle seems to be only half the story. Nutrients go in and waste comes out. This is a throughput, not a cycle. Also the diagram does not indicate whether the ribozymes are made by the cycle. I think there is some room for improvement in this diagram.

Fig 8 is very reminiscent of Fig 1 of Wu and Higgs (2009) J Mol Evol 69:541-554 and Fig 1 of Wu and Higgs (2011) Astrobiology 11:895-906.

Once again in Fig 8 I think that the cycle is over-emphasized and the throughput is not included. For example there must be ways of making nucleotides from scratch. It cannot be true that the only source of nucleotides is by degrading oligomers.

Section 3.4 - 'There is a debate whether CMIO could have formed by a CCSI consisting entirely or mainly of RNA'. This section seems to mix up two important questions. (i) Are RNA catalysts sufficient? Do we need other kinds of biomolecules like protein catalysts? (ii) Do we need the RNA system to be enclosed in a cell membrane or other kind of compartment? If so, can RNA control the growth and division of the membrane? These issues need to be considered separately.

line 554 - The case that two genes were essential initially is not made. Can we not have an RNA polymerase that uses other copies of itself as a template? If chemistry supplies monomers, then only one gene is necessary. The input and output would not necessarily need to be controlled by another gene. It would seem difficult to evolve two separate ribozyme functions at the same time.

If there were two genes, what were the functions of the two genes? The connection between the reactions catalyzed by the ribozymes and the concepts of CCSI and CMIO is not very clear at this point.

The conclusion comes back to the point about primitive Earth environment. This links to the introductory section about hydrothermal conditions etc. I agree that the question of the relationship between RNA World and the environment is important, but there is nothing in the main part of this article that addresses this question. It is only mentioned in the introduction and conclusions.

In summary, I think there are a lot of interesting points here, but the article could use some streamlining to emphasize the key new ideas.

Reviewer 2 Report

According to the abstract, this manuscript covers a number of different areas relating to the RNA world hypothesis, including its compatibility with (likely) primitive Earth conditions; the likely characteristics of an RNA world; and, finally, the relationship between an early life-like RNA world system(s) and other life-like systems. From the above, the manuscript then proposes that the first RNA world system may have consisted of two RNA genes of 100 base pairs each (this would probably be better described as genes of 100 nucleotides, unless the author is really proposing the presence of double–stranded RNA?). The author appears to have an (applied?) chemistry background (and has published a number of physical chemistry studies investigating the compatibility of the RNA world hypothesis with a hydrothermal vent origin of life), but has also published extensively in the social sciences area, for example applying evolutionary concepts to other 'life-like' systems, such as social insect colonies and human civilizations. There are a number of interesting points raised along the way, such as the generally one-to-one correspondence between DNA gene -> mRNA transcript -> protein enzyme -> single reaction (although many enzymes are often involved in a single pathway which transforms/produces a single substrate/product). Another interesting idea that is also raised is the possible importance of RNA solubility at high temperatures, though it does seem that the author is battling nobly (if possibly in vain) to reconcile the RNA world hypothesis with a hydrothermal vent origin of life scenario.

However, I find the manuscript contains a number of weaknesses:

One of the main issues is that paper suffers from trying to cover too many ideas, and because of this is quite long and unfocused. It is also fairly rambling: for example, lines 95-106 could be removed,

The second point is that the paper contains too many errors. One example obvious to me is the reference to one of my papers [ref. 11] which states that "chemical reactions catalyzed by protein-like molecules could form ambiguous information by...coevolution with RNA", which is not at all what is written in the paper.

The English is a problem too: "soft molecules" is not a typical way of describing organic molecules, and "formation and deformation" of molecules would be better expressed as formation and degradation. Statements such as in line 232: "the first step is to limit the characteristics of life-like systems into life on Earth" leave one scratching ones head.

More fundamentally perhaps, the distinction that is made between CCSI and CMIO is, as the author himself notes, remarkably similar to that between genotype and phenotype, and information and function, and even replication and metabolism. The finding that two RNA molecules would have been sufficient to start life, seems a little like the idea that has been proposed of an RNA template and its complementary copy originally functioning as an informational/functional(= ribozymic) pair (see for example: Shay JA, Huynh C, Higgs PG (2015) J Theor Biol. 364:249-59). Also, as far as this reviewer is aware, there are no known general (= nonspecific) RNA replicase ribozymes consisting of only 100 nucleotides.
